# Modular nonlinear hybrid plasmonic circuit

Alessandro Tuniz (ID) [1,2✉], Oliver Bickerton[1], Fernando J. Diaz[1], Thomas Käsebier[3], Ernst-Bernhard Kley[3], Stefanie Kroker[4,5], Stefano Palomba (ID) [1,2] & C. Martijn de Sterke (ID) [1,2]

Photonic integrated circuits (PICs) are revolutionizing nanotechnology, with far-reaching applications in telecommunications, molecular sensing, and quantum information. PIC designs rely on mature nanofabrication processes and readily available and optimised photonic components (gratings, splitters, couplers). Hybrid plasmonic elements can enhance PIC functionality (e.g., wavelength-scale polarization rotation, nanoscale optical volumes, and enhanced nonlinearities), but most PIC-compatible designs use single plasmonic elements, with more complex circuits typically requiring ab initio designs. Here we demonstrate a modular approach to post-processes off-the-shelf silicon-on-insulator (SOI) waveguides into hybrid plasmonic integrated circuits. These consist of a plasmonic rotator and a nanofocusser, which generate the second harmonic frequency of the incoming light. We characterize each component's performance on the SOI waveguide, experimentally demonstrating intensity enhancements of more than 200 in an inferred mode area of 100 nm$^2$, at a pump wavelength of 1320 nm. This modular approach to plasmonic circuitry makes the applications of this technology more practical.

[1] Institute of Photonics and Optical Science, School of Physics, The University of Sydney, Sydney, NSW 2006, Australia. [2] The University of Sydney Nano Institute, The University of Sydney, Sydney, NSW 2006, Australia. [3] Institute of Applied Physics, Abbe Center of Photonics, Friedrich Schiller Universität Jena, Max-Wien-Platz 1, 07743 Jena, Germany. [4] Physikalisch-Technische Bundesanstalt, Bundesallee 100, 38116 Braunschweig, Germany. [5] LENA Laboratory for Emerging Nanometrology, Technische Universität Braunschweig, Pockelsstrasse 14, 38106 Braunschweig, Germany. ✉email: alessandro.tuniz@sydney.edu.au

Chip-based nanophotonic waveguides that incorporate photonic and electronic functionality on a compact, monolithic platform[1] promise to revolutionize communications, sensing, and metrology[2–4]. The most promising approach being pursued relies on expanding existing silicon-on-insulator (SOI) technologies from the electronic to the optical domain, to produce photonic integrated circuits (PICs) exhibiting superior performance in terms of bandwidth and speed[5,6]. The quest for optical miniaturization is ultimately limited by diffraction—which in silicon corresponds to a maximum achievable spatial confinement of approximately 200 nm at telecommunication wavelengths. One of the most promising approaches for overcoming the diffraction limit by several orders of magnitude relies on nano-plasmonic structures[7], which harness metals to compress light down to the molecular, and even atomic scale[8,9]. Moreover, the giant intensity enhancement provided by plasmonic nanofocusing—typically ~100–2000 times[10]—has attracted interest for ultrafast, high-bandwidth, low-power nonlinear optics applications[11,12], e.g., for nano-scale sensing[13] and all-optical wavelength conversion[14]. Plasmonics can be harnessed for nanoscale second- and third-harmonic generation, which respectively relied on either the large surface $\chi^{(2)}$ or bulk $\chi^{(3)}$ of the metal itself[15–17], or on the large intensity enhancement within a dielectric at a plasmonic nanofocus[14]. This has mainly been demonstrated in planar structures that cannot be efficiently interfaced to PICs[18].

Interfacing waveguide-based PICs with plasmonic nanostructures is challenging: typically, the latter is hindered by large losses (due to metallic absorption) and low coupling efficiency (due to extreme differences in the participating mode profiles). PICs and plasmonics can be married using hybrid plasmonic waveguides (HPWGs) containing a low-index buffer layer between the metal and the high-index waveguide, enabling relatively low propagation loss without sacrificing plasmonic confinement, and providing a convenient intermediate interface for coupling between photonic and plasmonic waveguides[19,20]. Whereas the efficient energy transfer between PIC-compatbile photonic and plasmonic structures has been under intense experimental investigation with a diverse range of functionalities[21–27], including HPWG experiments demonstrating tight confinement and low propagation losses[28–30], nonlinear experiments using this platform have been limited[31].

While a number of simple HPWGs have been reported, the next challenge is to incorporate them into a more complex circuit with multiple modular, functional elements[32]—analogously to conventional PICs[1]. Ideally, such structures would be entirely chip-based, and be accessible using standard, industry-norm photonic components, thus simplifying the integration with more conventional technologies. Here we present the design, fabrication, and characterization of such a circuit, operating at $\lambda = 1.32\ \mu m$. It consists of two modules: a mode converter that efficiently transforms an incoming photonic transverse electric (TE) mode into a hybrid-plasmonic transverse magnetic (TM) mode, followed by a plasmonic nanofocuser that functions as a nonlinear wavelength converter. We note that standard solutions exist for the coupling of light into the TE photonic waveguide, which here is achieved by using a grating with an incident free-space Gaussian beam. In this way, our device thus represents a fully integrated chip by which a free-space Gaussian beam is focused to a cross-section that is almost two orders of magnitude below the diffraction limit in silicon, with a concomitant increase in intensity. To demonstrate that this increased intensity is due to the focuser, we fabricate and characterize two similar devices: one with a partial focuser and one with no focusing element at all. Note that while preliminary reports of both a TE-to-TM rotator[33] and directional-coupling-based TM-nano-focuser[30] have been reported separately, this is a proposal and demonstration of combining these two modular elements into a monolithic PIC-compatible plasmonic integrated circuit. This approach has clear advantages in terms of both design flexibility (enabling an industry-standard TE-waveguide input to achieve plasmonic nano-focusing), and wider bandwidth (enabled by the quasi-adiabatic modal evolution).

## Results

**Circuit design.** Our on-chip hybrid plasmonic integrated circuit (HPIC) is formed by two in-series plasmonic elements on a SOI waveguide (WG): a mode converter and a focuser. The latter combines a taper and a sharp tip, which functions as a nonlinear nanoscale light source. In our particular demonstration, we probe second harmonic generation (SHG) in the visible from a near-infrared pump. Figure 1a shows a schematic of the HPIC. The first component (i) is formed by a polarization rotator[33] (also operating as a TE-photonic to TM-plasmonic mode converter[34]); the second (ii) is a nanofocusing gold tip[10] resulting in SHG due to the intense nanoscale localization of the optical field, combined with the large surface $\chi^{(2)}$ of gold[18]. Figure 1c shows an electron micrograph of a fabricated HPIC on a SOI waveguide, highlighting the ~10 nm tip sharpness, which is limited only by the gold grains generated during the evaporation process[35].

To analyze our circuit we first consider the relevant HPIC modes during propagation. Figure 2a shows the result of 2D finite element (FE) simulations (COMSOL) of the modal evolution along the HPIC. Figure 2a also shows a top-view schematic of Fig. 1 for clarity. In the first instance, a gold film[36] ($t_{Au} = 50$ nm) with a SiO$_2$ spacer underneath[37] ($t_{spacer} = 20$ nm) gradually extends on a silicon waveguide (350 nm × 220 nm, $n_{Si} = 3.5$) until complete coverage (here, $\ell_{strip} = 30$–300 nm, as defined in Fig. 2). The red line in Figure 2a shows how the hybrid-TE (HTE) mode evolves within the waveguide, in terms of the real effective index and loss. The input is the fundamental TE-SOI mode of the bare waveguide, which excites the HTE mode (i) that rotates into a hybrid-TM mode (HTM) (ii). The HTM mode is then converted to a deep-subwavelength HTM plasmonic mode (iii)[38] by reducing the gold strip width ($w_{strip} = 300$–10 nm, as defined in Fig. 2). The z-component of the time-averaged Poynting vector $S_z$ associated with each participating mode is shown in Fig. 2b, and presents the salient features of the evolution of TE-SOI mode after it couples to the HTE mode. The modal evolution of the equivalent HTM mode is shown as the blue curve in Fig. 2a for completeness. The TE-SOI waveguide mode excites both the HTE and HTM hybrid plasmonic eigenmodes in location (i), each evolving in a non-trivial way along the device.

We next calculate the performance of the full device using full 3D FE simulations. Due to the many parameters, materials, and functionalities involved, the optimization of the complete device is challenging: first, a suitable compromise between adiabaticity (requiring a slow modal transition, i.e., a long device length) and loss (requiring short device lengths) is required; secondly, small changes in geometric parameters, alignment, and surface roughness can have a significant impact on the conversion efficiency. However, this process can be significantly simplified by using the modularity, which enables us to consider each element separately.

We model the fabricated structure shown in Fig. 1c. The cross-section of the $E_x$ and $E_y$ field components in the middle of the Si-WG are shown in Fig. 3a. Note in particular the polarization rotation in the spacer, manifesting as a vanishing $E_x$ component and an emerging $E_y$ component. A detailed plot of the electric field intensity $|\mathbf{E}|^2$ within the spacer near the tip is shown in Fig. 3b, showing a strong local enhancement at the tip apex. We calculate a ~1200× intensity enhancement at the gold

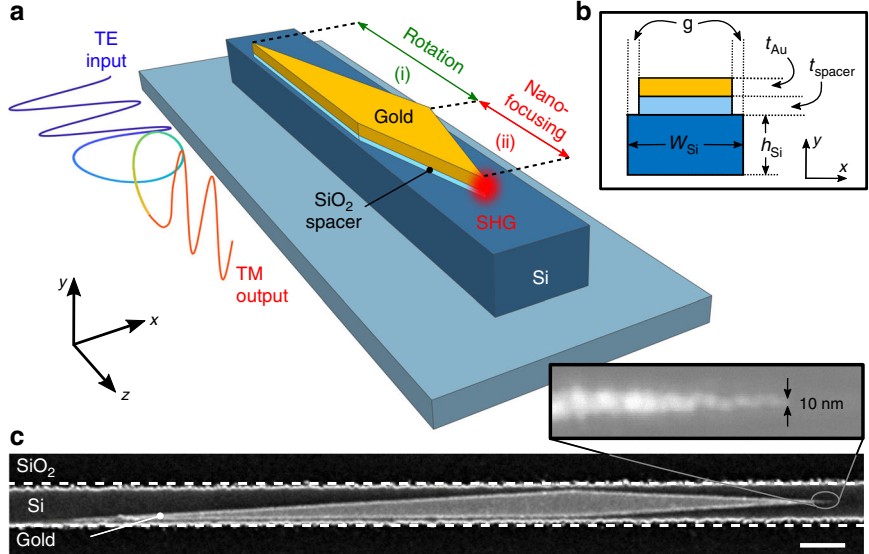

**Fig. 1 Schematic and scanning electron micrograph of silicon-on-insulator hybrid plasmonic circuit. a** An industry-standard TE ridge waveguide is followed by two in-series plasmonic circuit modules: (i) efficient TE-photonic to TM-plasmonic rotator and (ii) nano-focusing tip. **b** Geometric parameters at the rotator-focuser boundary. **c** Scanning electron micrograph (SEM) of a fabricated device. Scale bar: 400 nm. Here, $w_{Si} = 350$ nm; $h_{Si} = 220$ nm; $g = 25$ nm; $t_{spacer} = 20$ nm; $t_{Au} = 50$ nm Inset: high-resolution nanotip detail, revealing 10 nm apex sharpness.

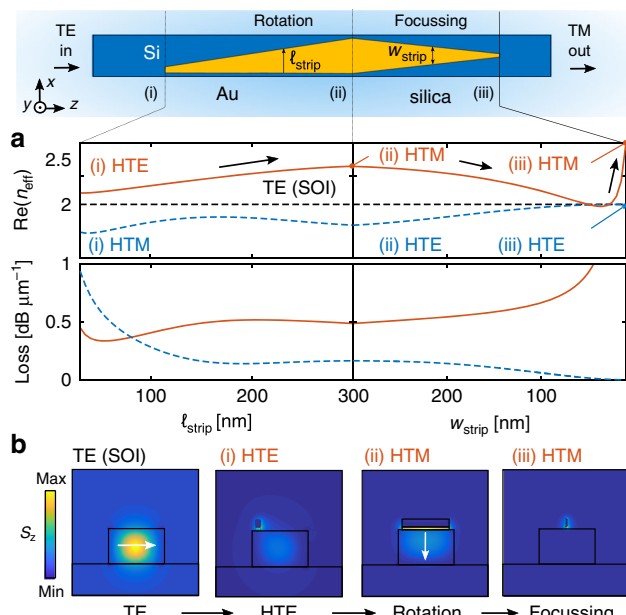

**Fig. 2 Two-dimensional calculations of relevant optical modes.**
**a** Calculated effective index $\mathbb{R}e(n_{eff})$ and loss for relevant modes ($\lambda = 1.32$ µm). Red line: a TE mode at input (TE SOI) (i) couples to a hybrid TE (HTE) mode, (ii) evolving to a rotated hybrid TM (HTM) mode, and (iii) transitions to a nano-focused HTM plasmonic mode. The dashed blue line shows the HTM-to-HTE evolution. **b** Corresponding mode profiles as labeled. White arrows represent the dominant electric field direction. The color represents the z-component of the time-averaged Poynting vector $S_z$. Window size: $0.8 \times 0.8$ µm². See Fig. 1 caption for relevant parameters.

surface with respect to the peak intensity in the silicon for the TE-SOI input. Figure 3c shows $S_z$ in each $xy$ cross-section as indicated by the dashed lines in Fig. 3a(i)–(iv). We calculate the conversion efficiency between the incoming TE-SOI mode and each of the participating modes in the full device by performing overlap integrals between the calculated 3D fields of Fig. 3c and the 2D modes of Fig. 2, as outlined in ref. [34].

The mechanism that converts the TE mode at input (Fig. 3a (input)) to the TM mode at the end of the rotator (Fig. 3a(ii)) is complicated by the fact that the waveguide evolves continuously over wavelength-scale propagation lengths, and that this is a lossy structure. In the rotator section, the gold-nanofilm overlayer tapers sideways relative to the underlying silicon waveguide, beginning on one corner on top of the silicon waveguide. Since the beginning of the rotator is formed by a sharp gold nanotip off-axis (Fig. 3a(i)), energy is distributed between the HTE and HTM mode, with most of the energy being coupled into the HTE mode for the TE SOI WG input considered. Our calculations using the method presented in ref. [34] indicate that the HTE and HTM modes at the start of the rotator are excited with a coupling efficiency of 70% and 30%, respectively (Fig. 3c(i)). As the gold film gradually tapers sideways to cover the waveguide, these two orthogonal modes evolve by rotating their polarization. In a quasi-adiabatic treatment[33], as the gold film gradually tapers sideways to cover the waveguide, the rotation mechanism can be interpreted to originate from the dominant electric field remaining orthogonal to the metal surface. Due to the asymmetry at input, the input HTE mode of the waveguide rotates into the HTM mode. A pioneering experimental study identified three possible regimes, depending on the rotator length chosen[33]: a non-adiabatic regime (short coupler, low power transfer); an adiabatic regime (long coupler, strong absorption); and a quasi-adiabatic regime with good power transfer to the desired mode at an intermediate length, which is the region where we operate. We obtain a TE-to-HTM (rotator) conversion efficiency of 41%, comparable to previous reports[33], and a TE-to-HTM (nanofocus) conversion efficiency of 12%, also comparable to the state of the art for plasmonic nanofocusing[14]. Note that 9% of the TE mode remains in the WG at output, which can be improved, for example, by more sophisticated multi-section rotator designs[39].

**Fabrication and linear experiments.** With an eye on the potential for modular approach to enhance off-the-shelf photonic waveguides with tailored plasmonic functionality, we purposefully choose to integrate our HPICs on previously fabricated SOI-WGs with standard electron-beam lithography and

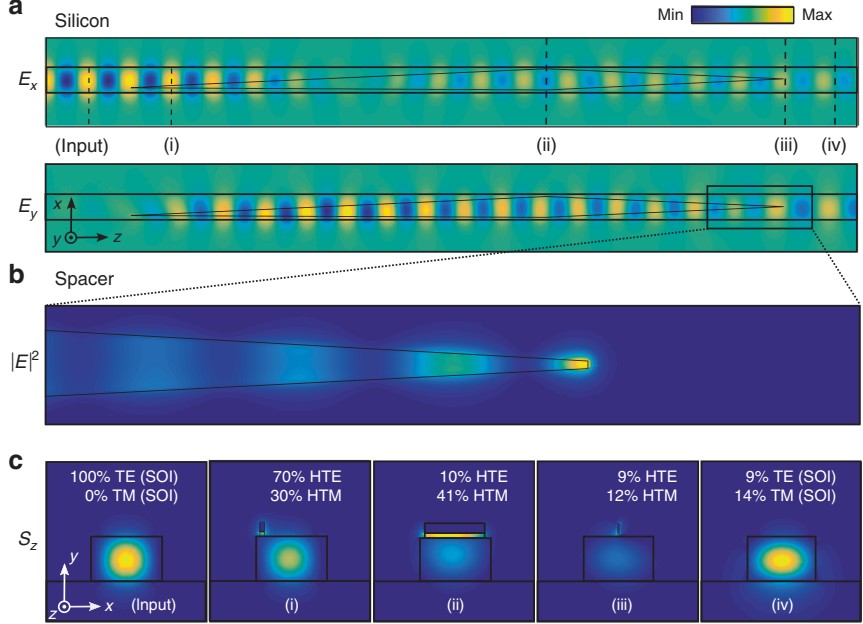

**Fig. 3 Three-dimensional finite element simulations showing device performance. a** Electric field components in the $xy$ plane (**a**) in the middle of the silicon waveguide, showing polarization rotation (vanishing $E_x$ and emerging $E_y$). Window size: $7 \times 1\,\mu m^2$. **b** Field intensity $|\mathbf{E}|^2$ in the middle of the spacer, showing nano-concentration of energy. Window size: $1.5 \times 0.2\,\mu m^2$. **c** Time-averaged Poynting vector $S_z$ at each of the locations (i)–(iv) and input as labeled in (**a**). Inset: relative contribution (in %) for each mode shown in Fig. 2[34].

evaporation techniques. Figure 4 shows an optical microscope image of an example bare SOI-WG with length $L = 20\,\mu m$: light is coupled in and out of the waveguide with shallow gratings optimized for TE polarization. For further details of the bare SOI-WG design and characterization, see "Methods" and Supplementary Fig. 1. The HPIC nanostructures, shown in Fig. 1c, were deposited on the WG in a subsequent step via combination of electron-beam lithography, SiO₂/gold evaporation, and lift-off—note in particular the excellent quality of the gold film, the sharp tip obtained, and the high alignment precision (<10 nm resolution). The details of the HPIC fabrication procedure are presented in the "Methods" and in Supplementary Fig. 2. Preliminary experimental waveguide characterization in the near-infrared (NIR) was performed by coupling light from free space ($\lambda = 1320\,\mu m$) onto the waveguide input grating coupler using a 100× near-infrared microscope objective (NA = 0.85—Olympus) and observing the light scattered by the device using a InGaAs camera (NIRvana, Princeton Instruments) (see "Methods" and Supplementary Fig. 3). The resulting measurement is shown in Fig. 4b. The field emerging from each nanoscale tip appears as a diffraction-limited spot, since all tips have physical dimensions well below the diffraction limit. We observe a diffraction-limited spot at the expected location of the gold nanotip, as well as residual TE light contained within the waveguide (in agreement with the simulations, see Fig. 3a(iv)), originating from the output grating. Figure 4c shows the same measurement when inserting a polarizer between the sample and camera with different orientations: we measure that the diffraction-limited spot is longitudinally (TM) polarized[40], confirming polarization rotation and that light exiting the grating is TE polarized. As further confirmation, Fig. 4d shows a direct comparison of the amount of light exiting the grating in the presence of the HPIC, with respect to an adjacent control sample without the HPIC. From the ratio of the total power scattered by each TE grating under comparable input conditions (see Supplementary Fig. 4), we conclude that the residual light in the TE waveguide in the presence of the HPIC, relative to the bare SOI waveguide, is $(13 \pm 1)\%$, in agreement with 3D simulations (see Fig. 3c(iv)).

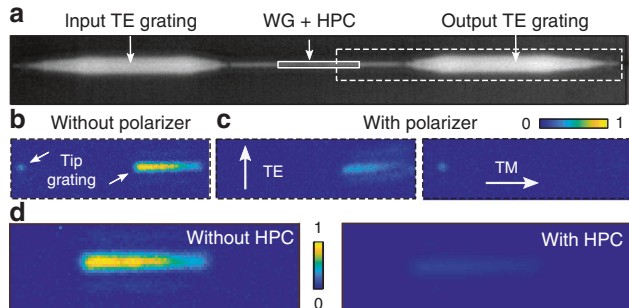

**Fig. 4 Measurements of polarization rotation. a** Microscope image of a 20 μm WG with an HPIC (SEM inset). Light is coupled to the waveguide using a TE grating at input. **b** Measured light from the output TE grating and the tip (dashed line in (**a**)). **c** Placing a polarizer between scattered light and camera confirms that light from the tip is TM polarized, while light from the grating is TE polarized. White arrows in (**c**) show polarizer orientation. **d** Measured light scattered by the output grating with and without the HPIC, resulting in 13% relative transmittance. Colorscale in (**b**), (**c**), and (**d**) represents the number of photon counts measured by the InGaAs camera, divided by the maximum number of counts measured in (**b**) and (**d**), respectively.

**Nanofocusing and nonlinear enhancement.** Plasmonic nanofocusing leads to spot sizes that are well below the diffraction limit, so that far-field linear optical experiments are inherently incapable of characterizing the focusing performance of our HPIC. Here we harness the high field intensities at the apex of the gold tip to estimate the field enhancement via nonlinear SHG experiments. Here, the surface nonlinear susceptibility $\chi^{(2)}$ of gold dominates over that of all surface and bulk sources of all constituent materials[41]. Ultrashort pump pulses ($\lambda_P = 1320$ nm, 200 fs, 80 MHz[31]) are coupled into the TE mode of the photonic waveguide via a grating coupler. They then enter one of three HPIC-enhanced WGs, each possessing incrementally sharper tips: the three HPIC considered here are shown in Fig. 5a.

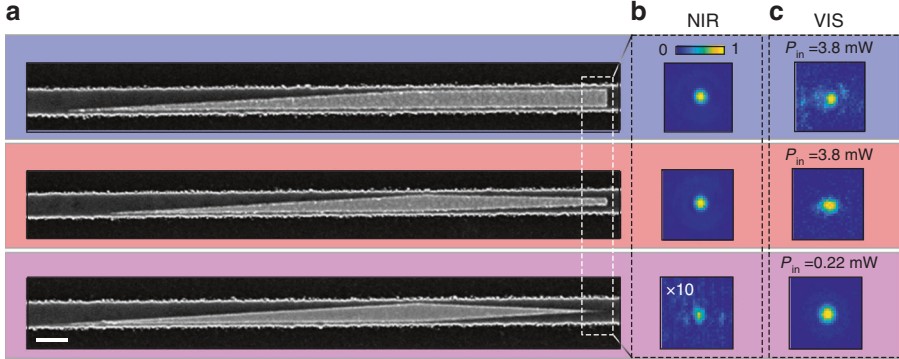

**Fig. 5 Measurements of pump- and second-harmonic light scattered by the gold nanotips. a** SOI-HPIC scanning electron micrographs, with tip width of 300 nm (blue), 138 nm (red), and 10 nm (magenta). Measured scattering from the SOI-HPIC **b** in the NIR **c** and visible. Images in (**b**) and (**c**) are respectively captured under the same conditions, unless otherwise indicated. $P_{in}$ is the average power incident onto the input grating for the three captured images in (**c**). The color bar represents the number of counts measured in (**b**) and (**c**), divided by the maximum number of counts in (**b**) and (**c**), respectively.

Scattered light images by each HPIC captured using near-infrared (NIR) and visible (VIS) cameras (PIXIS—Princeton Instruments) are shown in Fig. 5b, c, respectively. While nonlinear generation/scattering occurs during propagation across the entire HPIC[15], due to the large absorption of silicon (approximately 12 dB over 10 μm at 660 nm[42]), the absence of phase matching, and the wavelength-scale propagation lengths considered, we can attribute the measured nonlinear signal only to the localized intensity at the edge of the gold tip from which the NIR light emerges. The spectra of the NIR pump and the visible radiation are shown in the inset of Fig. 6a. The figure confirms that the visible radiation indeed is the second harmonic of the pump since $\lambda_{SHG} = \lambda_p/2 = 660$ nm. We observe that the sharpest tip causes the least amount of NIR scattering, consistent with 3D simulations (see Supplementary Fig. 5). In contrast, this tip also causes the strongest visible light emission (see Fig. 5b, c, magenta), even though the incident power is an order of magnitude smaller than in the other two cases—a preliminary indicator of nonlinear enhancement. In this case, the input power is reduced by 10 times in order to avoid damaging the sharp due to the high field strength.

To quantify the nonlinear response of each tip, we measure the raw spectral yield versus incident power at the SHG wavelength, as shown in Fig. 6a (circles). The linear relationship between the square root of the yield and the average power incident on the sample $P_{in}$ (corresponding to a quadratic input power dependence, $I_{SHG}^{1/2} \propto P_{in}$), further confirms the mechanism of SHG. As a first conclusion, we note the dramatic increase in SHG intensity for the sharpest tip, which indicates that nano-focusing was achieved. We compare the slopes of the three curves quantitatively via a linear fit to the experiment, as shown in the dashed lines of Fig. 6a, and infer the relative intensity enhancement with respect to the strip. The results are summarized in Fig. 6b, which shows the intensity enhancement as a function of the tip width obtained using different approaches. Black crosses show the measured enhancement, obtained by taking the square of the slopes in Fig. 6a, normalized to $w_{strip} = 300$ nm. We experimentally observe a maximum intensity enhancement of a factor ~216 ± 16 for the sharpest gold tip relative to the gold strip (uncertainties are obtained from the confidence intervals of the straight line slopes in Fig. 6a). The predicted range of theoretical enhancement at the tip is shown in the dark blue shaded region of Fig. 6b (left axis), and was calculated using an Eikonal approach[38], in excellent agreement with both the experiment and the range of intensity enhancements at the tip predicted by full 3D simulations (light blue region—see Supplementary Fig. 5 for further details). Note that the enhancement predicted by the

Eikonal approach is inversely proportional to the effective mode area $A_{eff}$, the definition of which can vary[12,38,43]. For completeness, Fig. 6b shows the range of inferred effective mode area for $w_{strip} = 10$ nm, i.e., $A_{eff} \sim 50$–$200$ nm[2 42] (black circles, in agreement with published calculations[38]).

Finally, we estimate the SHG conversion efficiency. After taking into account the effect of all optical elements, we conclude that the maximum SHG power is emitted by the sample for the sharpest nanotip (Fig. 6a, magenta) is 2.3 fW for an incident power of 0.22 mW, corresponding to a net conversion efficiency of $10^{-11}$. Taking into account the coupling efficiency into the waveguide (14%, see Supplementary Fig. 1d), this corresponds to $\sim 0.7 \times 10^{-10}$ of the power in the waveguide before the plasmonic rotator, and $\sim 0.6 \times 10^{-9}$ of the inferred power in the TM mode at the tip (cfr. Fig. 2d(ii)). Though these values are comparable to optimized nonlinear plasmonic SHG geometries[44,18], our geometry has the significant advantage of being on a PIC-compatible platform. It is worth noting that only ~0.06% of the power generated by a TM point source on the surface of a silicon waveguide radiates upwards, whereas the great majority of the SHG light is scattered into (and absorbed by) the silicon waveguide (see Supplementary Fig. 6). Future work will focus on new strategies to make use of the generated SHG, e.g., using hydrogenated amorphous silicon with low absorption at visible wavelengths, which will enable measurements of the SHG signal captured by the photonic waveguide[45].

## Discussion

The conversion efficiency could be further improved by optimizing the individual modular elements. Separate calculations for each module predict a peak rotator conversion efficiency of 58% for a rotator length of 4 μm, and of 34% for a focuser length of 1 μm (keeping all other parameters constant), resulting in a compound conversion efficiency of 20%. This is in good agreement with equivalent calculations for the full device, predicting a maximum conversion efficiency of 24% for the same rotator and focuser lengths of 4 and 1 μm, respectively. Thus, we estimate that through modest changes of the device parameters (e.g., increasing the gold thickness or with multi-section tapers[39] with up to 95% conversion efficiency), the pump TE-to-TM efficiency could be improved by approximately 9×, which would lead to a ~80-fold increase in nonlinear conversion efficiency. Further improvements may be achieved either by incorporating 2D materials on the waveguide surface, which possess a $\chi^{(2)}$ that is at least 1 order of magnitude greater than gold surfaces[45]. Additionally, $\chi^{(3)}$ nonlinear effects such as third-harmonic generation and four-wave-mixing may be

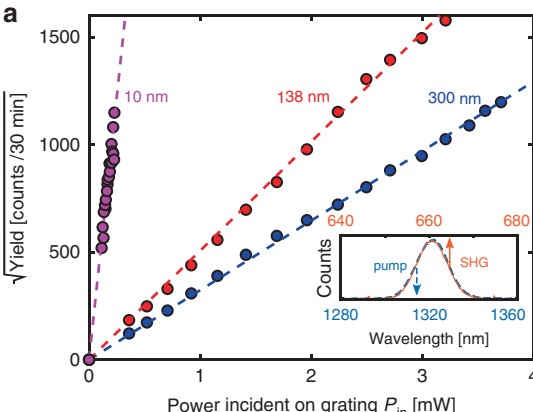

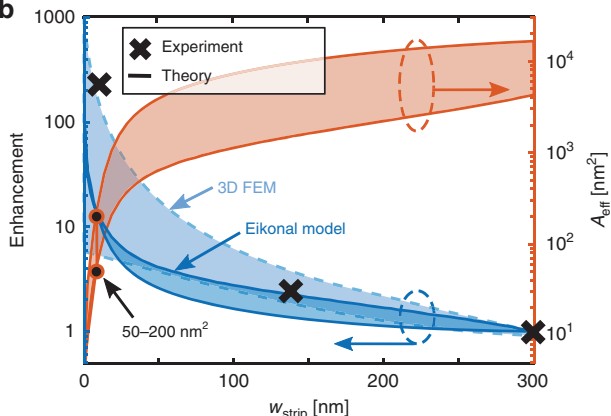

**Fig. 6 Experimental demonstration of nanoscale intensity enhancement in the HPIC. a** Circles: square root of the measured yield for each sample (color coding as in Fig. 5). Dashed lines: linear fits confirming quadratic dependence on incident power ($I_{SHG}^{1/2} \propto P_{in}$). Inset: spectra of the pump (blue), and of the SHG from the tip (orange). **b** Calculated enhancement (left axis, solid line) and effective area (right axis, dashed line) as a function of strip width for a focuser length of 3 μm, relative to the largest $w_{strip} = 300$ nm, following[38]. Light- and dark-blue shadow encompass the enhancement values predicted by full 3D simulations and the Eikonal model respectively, see Supplementary Fig. 5b. Black crosses show the experimentally measured relative increase in intensity, obtained from the square of the slopes in (**a**)—error bars from confidence intervals of the straight line fits in (**a**) are smaller than symbol size. Black circles: calculated effective area range of 50– 200 nm² for $w_{strip} = 10$ nm.

accessed by placing highly nonlinear materials at the nanofocus[14]. Further enhancement may be achieved with additional plasmonic modules, such as a bowtie nanoantenna[46] adjacent to the tip, or additional focuser and rotator modules which couple light back into the photonic waveguide.

This experiment represents a PIC-compatible, integrated nonlinear-plasmonic-SHG nanoscale light source, that makes use of two, in-series hybrid-plasmonic circuit elements. This design, fabrication, and characterization represent a TM plasmonic nano-focuser that is monolithically interfaced with an industry-standard TE-input SOI waveguide, and which can be coupled into by a conventional grating coupler. This work opens the door to the development of modular plasmonic circuit elements that can be seamlessly integrated on off-the-shelf photonic waveguides. Note that there has been a recent discussion of CMOS-compatible hybrid plasmonic waveguides[47,48], which requires using aluminum or copper as metals. We believe that this PIC can be easily fabricated using CMOS-compatible metals such as Cu and Al, which would

result in rotator modules with comparable TE-to-TM conversion efficiencies[34], as well as focuser modules with similar enhancement[38] (see Supplementary Fig. 7). Our approach unifies the emerging modular nanophotonic-circuit paradigm[32] with hybrid-integration of plasmonic nano-elements on industry-standard waveguides[27,26], extending the range of accessible structures to efficient hybrid plasmonic waveguides culminating in deep-subwavelength mode volumes, by performing three difficult-to-access optical functions (namely rotation, nano-focusing, and nonlinear conversion) back-to-back and on an integrated platform. This approach will facilitate access to efficient PIC-compatible deep-subwavelength field enhancements for on-chip quantum photonics and spectroscopy[49], nonlinear[13] and atomic-scale[9] sensing, and nanoscale terahertz sources and detectors[50].

## Methods

**Photonic waveguide grating design and characterization.** The waveguide gratings were designed in-house using a 2D solver CAMFR[51], with infinite air cladding and silicon substrate layer, a box layer of 2 μm thick, and a silicon waveguide layer of 220 nm, presenting grooves with an etching depth of $h_e$ and a period of $\Lambda$. Here, $h_e = 80$ nm and period $\Lambda = 440$ nm, resulting in a high coupling efficiency ($T_{up} = 51\%$), and wide bandwidth centered in $\lambda = 1320$ nm, low reflection ($R = 3.5\%$), and a selective in-coupling angle ($-11°$). From images of the optimized coupling to the waveguide, referenced to a mirror, we obtain a grating coupling efficiency of 14%, assuming that the loss due to each grating is equal. Waveguide losses without the HPIC are measured to be 0.12 dB μm⁻¹ using waveguides of different lengths. See Supplementary Fig. 1 for further details of the calculations, the calculated bandwidth, and experimental measurements of coupling- and propagation-losses.

**Hybrid plasmonic integrated circuit fabrication.** The plasmonic HPICs are integrated on the SOI waveguides as follows. First, the silicon waveguides are spin-coated with polymethyl methacrylate resist, and the HPIC structures are written with standard electron-beam lithography and developed with methyl isobutyl ketone. 20 nm silica and 50 nm gold are subsequently coated with electron-beam evaporation. Finally, a lift-off step (methyl-isobutyl-ketone) removes the resist. The alignment precision (~10 nm) is obtained using local gold markers, placed in the immediate vicinity of our off-the-shelf waveguides. See Supplementary Fig. 2 for a schematic of the fabrication procedure and alignment markers used.

**Experimental setup.** A detailed schematic of the experimental setup is shown in Supplementary Fig. 3. The source is an optical parametric oscillator ($\lambda_p = 1320$ nm, FWHM: 200 fs; repetition rate: 80 MHz). The power incident on the sample is controlled via a motorized half-waveplate placed before a polarizer. The beam is spatially shaped using a beam expander, telescope, and elliptical lens, so that its profile matches that of the input waveguide grating. A beamsplitter ($BS_{PM}$) and powermeter (PM) monitor the input power. A microscope holds the WGs and HPICs. Light is delivered and collected to the sample via a 100× NIR microscope objective (Olympus, NA = 0.85) and BS. A short-pass filter (850 nm) is included in SHG experiments to filter out the NIR light. The scattered light is measured with an imaging spectrometer, using NIR (NIRvana) and VIS (PIXIS) cameras. An additional NIR camera at a second output monitors alignment. The laser power drift is <±0.5%. Sample-to-sample waveguide coupling conditions fluctuate by ±4%, as obtained from the standard deviation of the total power emitted by the output grating of 10 nominally identical bare waveguide samples for optimized coupling conditions.

## Data availability
The data that support the findings of this study are available from the corresponding author upon reasonable request.

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

## Acknowledgements

A.T. acknowledges support from the University of Sydney Fellowship Scheme. S.K. acknowledges the support by the Deutsche Forschungsgemeinschaft (DFG, German Research Foundation) under Germany's Excellence Strategy—EXC-2123 Quantum-Frontiers—390837967. This work was performed in part at the NSW node of the Australian National Fabrication Facility (ANFF).

## Author contributions

A.T. conceived the idea and designed the experiment with input from S.P. and C.M.d.S. A.T. performed the simulations, experiments, and fabrication of the hybrid plasmonic circuit. O.B. designed the plasmonic device and fabricated the alignment markers. F.J.D. designed the bare silicon waveguides and the experimental setup. T.K., S.K., and E.-B.K. fabricated the bare waveguides. A.T. and C.M.d.S. wrote the manuscript with input from S.P. A.T., C.M.d.S., and S.P. directed the project.

## Competing interests

The authors declare no competing interests.
