## [Peer Review File · Nature Communications]

Reviewers' comments:

Reviewer #1 (Remarks to the Author):

The manuscript presents a nanophotonic platform, said to be CMOS compatible with the abilities to convert TE mode to TM, nanofocus it to $\sim 50\text{nm}^2$ and generate enhanced second harmonic. The authors justify their novelty in the first demonstration of doing all three together in an integrated device.

The results shown therein are sound and are in line with most of the claims raised by the authors. Namely, the Te-TM mode conversion, the signature of nanofocusing and enhanced SHG are all convincing. The fabrication is superb and so is the experimental skills. It is up to the editor to decide whether showing a combination of previously demonstrated effects merits publication in Nat. Comm. It might do.

I have some minor comments to the text:

1. Gold and silver, which are used for plasmonics are not really CMOS compatible.
2. The mechanism that converts TE to TM should be elaborated.
3. What is the origin of the optical nonlinearity? Surface susceptibility of the metal? The Si? It is not clear from the manuscript
4. In relating to figure 4, the authors claim to observe a "diffraction limited nanospot". I find this description quite awkward as nanophotonics aims to go beyond the diffraction limit (which is indeed 100-s of nanometers). I suggest dropping the "nano"

To summarize, I do not have any technical comments against the manuscript. It is just that the main claim is not very strong to my opinion but I do not object publication

I. Reviewer Requests:

R1. Gold and silver, which are used for plasmonics are not really CMOS compatible.

We used gold in our experiments since it is the preferred material in plasmonics, mainly for its ease of fabrication along with a high plasmonic quality coefficient [G. Li et al, ACS Photonics **5**, 1034-1040 (2018)]. Although we never mentioned CMOS in our original submission, we believe that considering the impact of using CMOS-compatible materials is important for future device designs. The performance of devices using copper or aluminium, both of which are CMOS compatible, is expected to be similar to that of the device described in our manuscript.

Rotator: The performance of the hybrid plasmonic rotator is weakly dependent on the choice of the metal chosen: Figure R1(a), adapted from [S. Kim and M. Qi, Optics Express **23**, 9968 (2015)] shows the calculated coupling efficiency (CF_{HP0}) from the TE mode to the TM mode as a function of device length (L_c) for copper (blue), gold (red), and aluminium (cyan), for a device that is very similar to ours. All experimental parameters for the structures used in Ref. [S. Kim and M. Qi, Optics Express **23**, 9968 (2015)] differ only slightly (less than 15%) from those considered in our manuscript.

Focuser: We have repeated the calculations to predict the relative enhancement as a function of the tip size (between 2-300 nm) using the simple model reported in [L. Lafone, et al., Optics Letters **39**, 4356 (2014)], considering experimental values for the relative permittivity ϵ_m of gold and copper (respectively $\epsilon_m = -79.986 + 7.1082i$ and $\epsilon_m = -79.738 + 9.8503i$ [P. B. Johnson and R. W. Christy, Phys. Rev. B **6**, 4370-4379 (1972)]) and aluminium ($\epsilon_m = -170.82 + 35.712i$ [A. D. Rakić, Appl. Opt. **34**, 4755-4767 (1995)]). Figure R1(b) shows the enhancement as a function of strip width for the three metals – all are quite similar and produce a relative enhancement $>100\times$ for a 10 nm tip.

Fig. R1. (a) Calculated coupling factor as a function of device length for the hybrid plasmonic polarization rotator with different metal films and waveguide width w_2 as labelled. The free space wavelength is $\lambda_0 = 1550\ \text{nm}$. Adapted from [S. Kim and M. Qi, Optics Express **23**, 9968 (2015).] (b) Calculated enhancement as a function of strip width for a focuser length of $3\ \mu\text{m}$ comparing different metals as labelled, with respect to the widest strip width considered ($w_{\text{strip}} = 300\ \text{nm}$), using the method outlined in [L. Lafone, et al., Optics Letters **39**, 4356 (2014)]. The free space wavelength is $\lambda_0 = 1320\ \text{nm}$. Calculations performed by Alessandro Tuniz.

The most important conclusion to draw from Fig. R1 is that both aluminum and copper, each of which are CMOS-compatible materials, provide comparable performance to that of gold. Thus, although the

specific device is not CMOS compatible, the underlying technology potentially is. Finally, note that recent reports have shown CMOS-compatible hybrid plasmonic waveguides using both copper [Dmitry Yu. Fedyanin et al, Ultralow-Loss CMOS Copper Plasmonic Waveguides, Nano Lett. **16**, 362-366 (2016)] and aluminium [G. Dabos et al., Scientific Reports **8**, 13380 (2018)] as metals. These two references have been included in the main manuscript as as Ref. [47] and Ref [48], respectively.

We have amended the manuscript as follows to reflect this new insight:

Note that there has been a recent discussion of CMOS compatible hybrid plasmonic waveguides [47, 48], which requires using aluminium or copper as metals. We believe that this PIC can be easily fabricated using CMOS-compatible metals such as Cu and Al, which would result in rotator modules with comparable TE-to-TM conversion efficiencies [34], as well as focuser modules with similar enhancement [38] (see Supplementary Fig. S7.)

We have included Fig. R1(b) in the Supplementary Information as Supplementary Fig. S7 to support this claim.

R2. The mechanism that converts TE to TM should be elaborated.

We thank the Reviewer for giving us the opportunity to elaborate on a very important aspect of this work.

The accepted physical interpretation for this hybrid plasmonic rotator device, as discussed in Ref. [J. N. Caspers, J. S. Aitchinson, and M. Mojahed, Opt. Lett. **38**, 4054-4057 (2013)], is that the dominant electric field stays orthogonal to the metal surface; thus, when the gold is tapered from the corner to cover the top of the waveguide, the input HTE mode of the waveguide is rotated into a HTM mode. We further note that, to the best of our knowledge, there has been no systematic study on the performance of lossy couplers. To address this request, three points should be considered:

(i) The coupling efficiency from the SOI input to the HTE and HTM modes at start of the rotator.

In the rotator section, the gold overlayer tapers sideways relative to the underlying silicon waveguide. At the beginning, the gold is located on one corner of the silicon waveguide (Fig. R2(i)): this sharp off-axis nanotip causes energy to be distributed between the quasi-orthogonal hybrid-TE and hybrid-TM mode, with the majority of the energy being coupled into the hybrid-TE mode, since the input of the waveguide is TE. Our follow-up calculations of the overlap integral between the 3D simulations (Fig. 3 of the manuscript) and the 2D simulations (Fig. 2) indicate that at the start of the rotator (Fig. 3(a)(i)) the HTE and HTM mode are excited with a coupling efficiency of 70% and 30%, respectively (now in Fig. 3(c)(i)).

(ii) The rotator evolves continuously over a wavelength-scale propagation length. As the film forming the rotator gradually tapers sideways and covers the silicon waveguide, the modes evolve in a non-trivial manner. Note that most of the literature on waveguide couplers considers lossless structures. In the presence of strong losses, the coupling behavior is significantly more complicated due to larger parameter space associated with complex modal Eigenvalues. In a lossless scenario, if one hybrid Eigenmode is excited at input and the waveguide properties change slowly upon propagation (i.e., the device length is much longer than the maximum local beat length of the Eigenmodes), then the coupler is classified as adiabatic, and a longer device improves the coupling. If instead multiple

modes are excited at input and the waveguide cross section does not change, then the coupler is classified as directional, with the maximum power transfer occurring at multiples of the half-beat length. The presence of losses implies that, for optimum performance, a compromise between the two must be reached, as we now discuss.

(iii) *The impact of losses on coupling.* A pioneering experimental study on a similar structure [J. N. Caspers, J. S. Aitchinson, and M. Mojahed, Opt. Lett. **38**, 4054-4057 (2013)] identified three regimes: 1) a nonadiabatic regime (short coupler, resulting in low power transfer), 2) a high loss regime (long coupler, resulting in absorption), and 3) an optimum at an intermediate length. Calculations show that in our device, we are close to this intermediate region: at the end of the rotator, the power of the TM mode is 41% of the bare waveguide input, and 10% in the TE mode, as described in the manuscript.

Considering the above, we have elaborated on the TE-to-TM coupling mechanism as follows:

The mechanism that converts the TE mode at input (Fig. 3(a)(input)) to the TM mode at the end of the rotator (Fig. 3(a)(ii)) is complicated by the fact that the waveguide evolves continuously for wavelength-scale propagation lengths, and that this is a lossy structure. In the rotator section, the gold-nanofilm overlayer tapers sideways relative to the underlying silicon waveguide, beginning on one corner on top of the silicon waveguide. Since the beginning of the rotator is formed by a sharp gold nanotip off-axis (Fig. 3(a)(i)), energy is distributed between the HTE and HTM mode, with most of the energy being coupled into the HTE mode for the TE SOI WG input considered. Our calculations using the method presented in Ref. [34] indicate that the HTE and HTM modes at the start of the rotator are excited with a coupling efficiency of 70% and 30%, respectively (Fig. 3(c)(i)). In a quasi-adiabatic treatment [33], as the gold film gradually tapers sideways to cover the waveguide the rotation mechanism can be interpreted to originate from the dominant electric field remaining orthogonal to the metal surface; due to the asymmetry at input, the input HTE mode of the waveguide rotates into the HTM mode. A pioneering experimental study identified three possible regimes, depending on the rotator length chosen [33]: a non-adiabatic regime (short coupler, low power transfer); an adiabatic regime (long coupler, strong absorption); and a quasi-adiabatic regime with good power transfer to the desired mode at an intermediate length, which is the region where we operate.

As a result, we have changed the term “adiabatic” in the final paragraph of the introduction to “quasi-adiabatic”, and expanded Fig. 3 of the manuscript to account for the above, as shown in Fig. R2 below.

FIG. 3. 3D FE simulations of device performance. (a) Electric field in the xy plane (a) in the middle of the silicon waveguide, showing polarization rotation. Window size: $7 \times 1 \mu\text{m}^2$. (b) Field intensity $|E|^2$ in the middle of the spacer, showing nano-concentration of energy. Window size: $1.5 \times 0.2 \mu\text{m}^2$. (c) Time-averaged power flow at each of the locations (i)–(iv) and input as labelled in (a). Inset: relative contribution (in %) for each mode shown in Fig. 2 [34].

Fig. R2. Amended Fig. 3 and caption in the main manuscript.

R3 What is the origin of the optical nonlinearity? Surface susceptibility of the metal? The Si?

The origin of the nonlinearity is indeed the surface nonlinear susceptibility of gold. Recent second harmonic generation experiments using near-infrared pump on gold and silicon films [F. Che et al, Results in Physics 7, 593–595 (2017)] – now Ref. [41] – showed that the second harmonic intensity in gold films was more than one order of magnitude larger than in silicon films. The manuscript has been amended as follows:

Here, the surface nonlinear susceptibility $\chi^{(2)}$ of gold dominates over that of all nonlinear surface and bulk sources of all constituent materials [41].

R4. In relating to figure 4, the authors claim to observe a “diffraction limited nanospot”. I find this description quite awkward as nanophotonics aims to go beyond the diffraction limit (which is indeed 100-s of nanometers). I suggest dropping the “nano”.

The wording in the original manuscript was imprecise. The relevant section has been amended as follows, and we have dropped the word “nano”, and added additional clarifying sentence (bold text):

The field emerging from each nanoscale tip appears as a diffraction limited spot, since all tips are well below the diffraction limit. We observe a diffraction-limited ~~nanospot~~ at the expected location of the gold nanotip [...]

II. Reviewer Comments

C1 [...] the TE-TM mode conversion, the signature of nanofocusing and enhanced SHG are all convincing. The fabrication is superb and so is the experimental skills.

We thank the Reviewer for acknowledging the high quality of the fabricated devices and experiments.

C2 The authors justify their novelty in the first demonstration of doing all three together in an integrated device. [...] It is up to the editor to decide whether showing a combination of previously demonstrated effects merits publication in Nat. Comm.

We take the opportunity to highlight that the recent scientific literature shows a marked increase in interest for two independent technological paradigms:

- 1) **Photonic modularization** (see for example J. Shi et al, “Modular assembly of optical nanocircuits”, Nature Communications **5**:3896 |(2014)). In that work, the authors point out that, although complex micro-electronic circuits are now ubiquitous, “*modern nanophotonic systems are still far from a similar level of sophistication, partially because of the lack of modularization of their response in terms of basic building blocks*”. That work pioneered nano-photonic circuit designs on the basis of nano-particles on planar substrates; here we extend that work to encompass photonic circuitry, and adding/combining modular components to the available nano-photonic component library, showing how they can be assembled in a compact device without sacrificing performance.
- 2) **On-chip hybrid integration of plasmonic elements**: see for example Z. Li, Nature Nanotechnology **12**, 675-683 (2017), and R. Guo et al., Science Advances **3**:e1700007 (2017). In those publications, the authors suggest that “*functional plasmonic elements [...] integrated with low-loss dielectric optical waveguides [...] create a fundamentally new link between electronic and photonic circuits.*” In that work however, only one linear, functionality at a time was shown (i.e., mode conversion, or signal routing), using wavelength-scale mode volumes inside the waveguide – our work expands on the concept, showing multiple back-to-back functions, culminating in a deep sub-wavelength mode volumes, and an enhanced nonlinear plasmonic performance.

To conclude: our work represents an important step in unifying and extending these two emerging nanotechnological paradigms (i) modularity, and (ii) hybrid plasmonic integration. We show the potential of the modular design for plasmonic waveguide systems on industry-standard photonic integrated circuits, demonstrating both efficient mode conversion and nonlinear optic functionality at the deep sub-wavelength scale, all on a convenient chip platform.

We have now included the above references (now Ref. [26], Ref. [27], and Ref. [32]) to the relevant portion of the introduction, and included a clarifying sentence further contextualizing the importance of this work in the concluding remarks:

Our approach unifies the emerging modular nanophotonic-circuit paradigm [32] with hybrid-integration of plasmonic nano-elements on industry-standard waveguides [26, 27], extending the range of accessible structures to efficient hybrid plasmonic waveguides culminating in deep-

subwavelength mode volumes, by performing three difficult-to-access optical functions (namely rotation, nano-focussing, and nonlinear conversion) back-to-back and on an integrated platform.

III. Additional Changes

Definition of the effective area: While revising the manuscript, it came to our attention that the definition of effective mode area (which in this manuscript we define as A_{eff}) is not unique. This is especially relevant in plasmonics, where metal corners generate local hot spots that affect the effective mode area. For example, Ref. [Optics Express **20**, pp. 11487-11495 (2012)] – now included as Ref. [43] – uses the definition:

$$A_m = \frac{\int_{-\infty}^{\infty} P(x, y) dx dy}{\max[P(x, y)]},$$

where P is the mode energy flux. However, Ref. [L. Lafone et al., Optics Letters **39**, 4356 (2014)] include an additional comment: “*Since the sharp corners of the metal film can exaggerate the confinement, we use the maximum electric energy density along the center of the waveguides in the denominator of Eq. (1)*”. Other choices, e.g., in nonlinear optics [G. Li et al., ACS Photonics **5**, 1034 (2018)], result from the choice of factorization following a first-principles derivation. Our previous effective area was quoted as 50 nm^2 for a tip sharpness of 10 nm. Further calculations show that it can be as high as 200 nm^2 – still ~ 2000 times below the diffraction limit mode area of $\lambda^2/4$, and comparable to the results in Fig. 4(a) of Ref. [L. Lafone et al., Optics Letters **39**, 4356 (2014)]. Since the definition of the effective mode area, and therefore of the relative intensity enhancement, is not unique, for completeness we have amended Fig. 6(b) of the manuscript and its caption to include the ranges of possible values, as shown in see Fig. R3 below.

We expanded the text in the main manuscript:

Note that the enhancement predicted by the Eikonal approach is inversely proportional to the effective mode area A_{eff} , the definition of which can vary [12, 38, 43]. For completeness, Fig. 6(b) shows the range of inferred effective mode area for $w_{\text{strip}} = 10 \text{ nm}$, i.e., $A_{\text{eff}} \sim 50\text{--}200 \text{ nm}^2$ [42] (black square circles, in agreement with published calculations [38]).

The abstract was also appropriately adapted:

[...] an intensity enhancement of >200 in a calculated mode area of **order 100 nm^2** [...]

FIG. 6. Experimental demonstration of nanoscale intensity enhancement in the HPIC. (a) Circles: square root of the measured yield for each sample (colour coding as in Fig. 5). Dashed lines: linear fits confirming quadratic dependence on incident power ($I_{\text{SHG}}^{1/2} \propto P_{\text{in}}$). Inset: spectra of the pump (blue), and of the SHG from the tip (orange). (b) Calculated enhancement (left axis, solid line) and effective area (right axis, dashed line) as a function of strip width for a focuser length of $3 \mu\text{m}$, relative to the largest $w_{\text{strip}} = 300 \text{ nm}$, following Ref. [38]. Light- and dark- blue shadow encompass the enhancement values predicted by full 3D simulations and the Eikonal model respectively, see Supplementary Fig. 5(b). Black crosses show the experimentally measured relative increase in intensity, obtained from the square of the slopes in (a) – errors bars from confidence intervals of the straight line fits in (a) are smaller than symbol size. Black square circles: calculated effective area range of $50\text{--}200 \text{ nm}^2$ for $w_{\text{strip}} = 10 \text{ nm}$.

Fig. R3. Amended Fig. 6 and caption in the main manuscript.

Estimation of fluctuations in coupling conditions:

Finally, we have further improved data presentation by including a discussion of potential sources of experimental error. We included two sentences to clarify the small ($\pm 4\%$) fluctuations of input coupling conditions into the waveguides, as well as the magnitude of the laser power drift ($\pm 0.5\%$) – see “Experimental setup” in the main manuscript:

The laser power drift is $< \pm 0.5\%$. Sample-to-sample waveguide coupling conditions fluctuate by $\pm 4\%$, as obtained from the standard deviation of the total power emitted by the output grating of 10 nominally identical bare waveguide samples for optimized coupling conditions.

We have also expanded the text estimating the experimentally measured plasmonic enhancement:

We experimentally observe a maximum intensity enhancement of a factor 216 ± 16 for the sharpest gold tip relative to the gold strip (uncertainties are obtained from the confidence intervals of the straight line slopes in Fig. 6(a)).

The caption of Fig. 6(b) has been expanded to include a clarifying comment in bold below:

Black crosses show the experimentally measured relative increase in intensity, obtained from the square of the slopes in (a) – errors bars from confidence intervals of the straight line fits in (a) are smaller than symbol size.

REVIEWERS' COMMENTS:

Reviewer #1 (Remarks to the Author):

The authors addressed adequately all my comments. I have no further remarks.

Reviewer comments:

The authors addressed adequately all my comments. I have no further remarks.

No action required.